# On Systematic Design of Fractional-Order Element Series

**DOI:** 10.3390/s21041203

**Published:** 2021-02-09

**Authors:** Jaroslav Koton, David Kubanek, Jan Dvorak, Norbert Herencsar

**Affiliations:** Department of Telecommunications, Faculty of Electrical Engineering and Communication, Brno University of Technology, Technicka 12, 616 00 Brno, Czech Republic; kubanek@vutbr.cz (D.K.); dvorakjan@vutbr.cz (J.D.); herencsn@vutbr.cz (N.H.)

**Keywords:** fractor, fractional-order element, generalized immittance converter, series design of fractors, “seed” FOE

## Abstract

In this paper a concept for the efficient design of a series of floating fractional-order elements (FOEs) is proposed. Using even single or a very limited number of so-called “seed” FOEs it is possible to obtain a wide set of new FOEs featuring fractional order α being in the range [−n,n], where *n* is an arbitrary integer number, and hence enables to overcome the lack of commercial unavailability of FOEs. The systematic design stems from the utilization of a general immittance converter (GIC), whereas the concept is further developed by proposing a general circuit structure of the GIC that employs operational transconductance amplifiers (OTAs) as active elements. To show the efficiency of the presented approach, the use of only up to two “seed” FOEs with a properly selected fractional order $\alpha_{\mathrm{seed}}$ as passive elements results in the design of a series of 51 FOEs with different α being in the range [−2,2] that may find their utilization in sensor applications and the design of analog signal processing blocks. Comprehensive analysis of the proposed GIC is given, whereas the effect of parasitic properties of the assumed active elements is determined and the optimization process described to improve the overall performance of the GIC. Using OTAs designed in 0.18 μm TSMC CMOS technology, Cadence Virtuoso post-layout simulation results of the GIC are presented that prove its operability, performance optimization, and robustness of the proposed design concept.

## 1. Introduction

In fractional calculus [1,2,3,4,5], generalizing derivatives with integer order to derivatives with non-integer or fractional order, has tremendously gained attention as it is applied in many and various engineering and research disciplines and areas, spanning biology [6,7,8,9], food [10,11,12], cybersecurity [13,14], modeling and control [15,16,17,18,19,20,21], signal processing [22,23,24], electrical engineering [25,26,27,28,29,30], and other. The reason for the increased interest in fractional-order calculus and system design may be seen in the fact that the presence of fractional order represents another degree of freedom to mathematically describe the behavior of a function block. This enables one to provide characteristics in between integer–orders in comparison to standard (integer-order) systems, which may become beneficial while more accurate signal generation and measurement, and/or system modeling and control is required.

Dealing mainly in the areas of signal processing, modeling and control, and electrical engineering, the implementation of required fractional-order function block relies on the presence of elements with fractional-order immittance, i.e., fractional-order elements (FOEs) or simply fractors. To design a FOE with required fractional order α (generally α∈IR), one of the direct implementations as recently summarized in [31] may be used, however all these techniques are still at the level of laboratory experiments. Hence, they do not provide readily available FOEs as discrete elements and mainly are suitable for capacitive FOE design only, i.e., 0<α<1. Additionally, the implementations as described in [31] enable one to obtain FOEs that are operable in a limited frequency band and with a narrow range of available α. To overcome the current obstacles in the unavailability of FOEs, they are commonly approximated by an RC network for the purpose of performance analysis and design verification by means of simulations or experimental measurements [32]. To approximate a FOE using an RC network, different approaches are described in the open literature, see e.g., [33,34,35]. However, for each different FOE, the RC network must be redesigned. This further limits the interest of the broader research community in fractional-order circuits and systems, as individual research groups use their “tuned” FOE that is mainly specified with its fix fractional order α.

To obtain FOEs featuring new values of fractional order α without re-designing the “tuned” FOE, the generalized immittance converter (GIC) may be efficiently utilized. Originally, the GIC was and still is used to emulate a classic inductor (and to obtain so called synthetic inductor) using resistors, capacitors, and selected types of active elements, e.g., operational amplifiers [36], current conveyors [37], current feedback operational amplifiers [38], etc. The utilization of GIC in designing factional-order elements was also discussed e.g., in [39,40,41,42,43], where Antoniou’s GIC employing operational amplifiers is used. The approach presented in [39] enables one to design new FOE with a fractional order between −2 and 2, but always requires a unique fractional-order element with specific α (i.e., 0.2, 0.3, 0.4, 0.5, 0.6, 0.7, 0.8). A comprehensive analysis of opamp-based Antoniou’s GIC, however limited to fractional-order inductor synthesis only, is provided in [40], similarly as in [41,42] the fractional-order inductor design and its utilization in the frequency filter design is discussed. First in [43], a more general approach to design FOEs is discussed, where both inductive and capacitive FOEs are used to design a set of new FOEs with a fractional order between −4 and 4. Note that as the Antoniou’s GIC is always used, the newly obtained FOEs are always grounded.

In this paper, we elaborate the efficient utilization of so called “seed” fractional-order elements featuring fractional order $\alpha_{\mathrm{seed}}$ that are employed in a general immittance converter to design a series of fractional-order elements. We partially presented this concept in [44], where the design of a series of grounded FOEs with fractional order [−2,2] was presented. Here, we further develop the theory and the design approach to obtain arbitrary and floating FOEs. The structure of the paper is as follows. In Section 2, the theory on fractional-order elements is shortly described. In Section 3, we present the concept of designing a FOE with fractional order α being from the arbitrary range [−n,n], where *n* is a positive integer number. Using operational transconductance amplifiers (OTAs) as active elements, we also propose possible implementation of the general immittance converter and use it to design a wide series of floating FOEs. In Section 4, the behavior of the proposed GIC is further analyzed. Taking into account the non-ideal behavior of the active elements, the design rules are discussed to optimize the overall performance of the GIC. Section 5 provides post-layout simulations of new FOEs obtained by employing the proposed GIC, whereas the optimization recommendations are also advantageously utilized to broaden the operational frequency band and increase dynamic. To show a practical utilization of the GIC and the design of fractional band-pass filter is also discussed as an example. Finally, Section 6 concludes this paper.

## 2. Theory on Fractional-Order Elements

Fractional-order elements are understood to be the simplest electrical elements whose impedance function follows fractional order differential equations and are used as basic building blocks for other fractional-order circuits and systems design. In the open literature, FOEs are also refereed to as constant-phase elements (CPEs) [45], elements with fractional impedance (EFIs) [46], or generally fractors [39] whose impedance in *s*-domain is defined as:(1)$$Z_{F}\left(s\right)=\frac{1}{s^{\alpha }\cdot F},$$
where *F*, called as fractance, is the coefficient of the fractor, and α is generally a real number, called the fractional-order.

In frequency domain, the magnitude of a fractor is $|Z_{F}|=1/(\omega^{\alpha }F$) Ω. For positive/negative values of α the magnitude $|Z_{F}|$ is monotonically decreasing/increasing with frequency by 20·α dBΩ/dec, whereas the phase angle remains always constant φ=−α·90 deg.

If 0<α<1, the phase angle of the fractor is negative and the fractor is called the fractional-order capacitor (also fractional capacitor, capacitive FOE, or capacitive fractor):(2)$$Z_{C\alpha }\left(s\right)=\frac{1}{s^{\alpha }\cdot C_{\alpha }},$$
where $C_{\alpha }=F$ is referred to as pseudo-capacitance or fractional capacitance.

For −1<α<0, the phase angle of the fractor is positive and the fractor is called the fractional-order inductor, also referred to as the fractional inductor, inductive FOE, or inductive fractor. The fractional order of the inductive fractor is commonly labeled as β, whereas it may be evident that β=−α:(3)$$Z_{L\beta }\left(s\right)=s^{\beta }\cdot L_{\beta },$$
where $L_{\beta }=1/F$ is the pseudo-inductance or fractional inductance.

As presented in [33,47], the fractional capacitor and its pseudo-capacitance $C_{\alpha }$ may be represented as the equivalent capacitor with capacitance *C* that features the same impedance at frequency $\omega_{0}$:(4)$$C=\frac{C_{\alpha }}{\omega_{0}^{1-\alpha }},$$
and similarly for the fractional inductor with its pseudo-inductance $L_{\beta }$, an equivalent inductor with its inductance *L* featuring the same impedance at frequency $\omega_{0}$ can be specified:(5)$$L=\frac{L_{\beta }}{\omega_{0}^{1-\beta }}.$$

It may be noted that for α=0,1,or−1, the fractor defined by (Equation 1) becomes resistor, capacitor, or inductor, respectively. For |α|>1, the fractor (Equation 1) can be used to describe higher-order immittances, e.g., the frequency dependent negative resistor (FDNR), finding their application in a higher-order frequency filter design [48,49].

## 3. General Immittance Converter in FOEs’ Series Design

As already discussed in Section 1, it is not necessary to limit the utilization of GIC to design synthetic inductors. The general immittance converter may also be efficiently used in fractional-order element design as shown e.g., in [39], where the known operational amplifier-based Antoniou’s GIC was employed. Here we further extend the idea of transforming FOEs and provide a concept of efficient design of a series in fractional order α of fractional-order elements by using even single or very a limited number of “seed” FOEs.

### 3.1. General Immittance Converter Behavior Definition

Assume a general function block as shown in Figure 1 that is represented by general active/passive network to which general admittances Y${}_{i}$ (i=1,…,n; *n* being even number) are connected. The general active/passive network may represent arbitrary interconnection of an arbitrary type of active and passive elements and is determined by its parameter *g*, a transcondustance specific for this active/passive network. Let the input admittance ($Y_{\mathrm{IN}}$) of such a general function block be defined as:(6)$$\left[Y_{\mathrm{IN}}\right]\;=\;\left[\begin{array}{cc} 1 & -1\\ -1 & 1 \end{array}\right]\frac{\prod_{i=1}^{n/2}Y_{\left(2i\right)}}{\prod_{i=1}^{n/2}Y_{\left(2i-1\right)}}g.$$

The general admittances Y${}_{i}$ (i=1,…,n) may be represented by any type of passive element, such as conductor (G), inductor (L), capacitor (C), or fractional-order element (FOE), whereas adopting the nomenclature as defined in Section 2, for each passive element, i.e., conductor, inductor, capacitor, and FOE it is possible to claim that its fractional order $\alpha_{i}$ equals to 0, −1, 1, and $\alpha_{\mathrm{FOE}}$ ($-1<\alpha_{\mathrm{FOE}}<0$ or $0<\alpha_{\mathrm{FOE}}<1$), respectively. Under these assumptions, for the fractional order α defining the phase angle of the input admittance (Equation 6) can be written:(7)$$\alpha =\sum_{i=1}^{n/2}\alpha_{\left(2i\right)}-\sum_{i=1}^{n/2}\alpha_{\left(2i-1\right)},$$
and the feasible range of fractional order α is defined as [−n,n].

To better demonstrate the advantageous features of the proposed concept of designing a series in fractional order α of fractional-order elements, let n=4. Then (Equation 6) and (Equation 7) simplify to:(8)$$\left[Y_{\mathrm{IN}}\right]\;=\;\left[\begin{array}{cc} 1 & -1\\ -1 & 1 \end{array}\right]\frac{Y_{2}Y_{4}}{Y_{1}Y_{3}}g,$$
and
(9)$$\alpha =\alpha_{2}+\alpha_{4}-\alpha_{1}-\alpha_{3},$$
respectively.

As in practical analog circuit design, classic inductors, and/or inductive fractors are not commonly used, in the further text it is assumed that the general admittances Y${}_{i}$ (i=1,…,4) may be replaced only by conductors ($\alpha_{i}=0$), capacitors ($\alpha_{i}=1$), and/or capacitive FOEs ($\alpha_{i}=\alpha_{\mathrm{FOE}}$, $0<\alpha_{\mathrm{FOE}}<1$). Now replacing the general admittance Y${}_{i}$ (i=1,…,4) by one of the three assumed types of passive elements, the following set of passive (synthetic) elements observed at the input of the immittance converter and specific with their fractional order α can be described:Frequency dependent negative resistor - type I (FDNR-I), α=2,Fractional FDNR-I, 1<α<2,Capacitor C, α=1,Capacitive FOE, 0<α<1,Resistor R, α=0,Inductive FOE, −1<α<0,Inductor L, α=−1Fractional frequency dependent negative resistor-type II (FDNR-II), −2<α<−1,FDNR-II, α=−2.

Note that the feasible range of fractional order α is now [−2,2] only, which is caused by the fact that neither classic nor fractional inductors are assumed to replace one or more general admittances Y${}_{i}$ (i=1,…,4).

The frequency dependent negative resistor-type I (FDNR-I) is also referred to as the *D* element (or double capacitor) and features purely real negative resistance that decreases in magnitude with increasing frequency [36], whereas FDNR-II also exhibits purely real negative resistance, however, its magnitude increases for increasing frequency. Additionally, comparing with [39], the inductive FOE, fractional FDNR-II, fractional FDNR-I, and capacitive FOE, may be referred to as Type-I fractor, Type-II fractor, Type-III fractor, and Type-IV fractor, respectively.

Using a general immittance converter allows one to obtain a wide series of new FOEs using a very limited set of “seed” FOEs and their fractional order $\alpha_{\mathrm{seed}}$. As an example, assume a “seed” FOE with its fractional order $\alpha_{\mathrm{seed}}=0.2$. Using always at most two identical “seed” FOEs and two capacitors together with conductors to replace external admittances Y${}_{i}$ (i=1,…,4) in (Equation 8), then according to (Equation 9) 19 unique values of fractional order α from the range [−2,2] are obtained. The specific combinations of external passive elements, i.e., of the conductors, capacitors, and “seed” FOEs, are listed in Table A1.

To better comprehend the advantage in utilizing “seed” FOEs, even 51 different values of fractional order α, still from the range [−2,2], can be obtained by assuming $\alpha_{\mathrm{seed}1}=0.25$ and $\alpha_{\mathrm{seed}2}=0.0625$. As a result, for each α, the input admittance $Y_{\mathrm{IN}}$ (Equation 8) features a phase angle from the range [−180,180] deg as illustrated in Figure 2. The specific combinations of external admittances types defined by their $\alpha_{i}$ is summarized in Table A2. Hence, it may be obvious that using a very limited set of “seed” FOEs, a broad series of new fractional order elements primarily with different fractional order α may be obtained. Furthermore, by adjusting the values of external capacitors (*C*), conductors (*G*), and most preferably also the transcondustance *g* of the active/passive network it is possible to obtain a generally arbitrary value of the fractance being observed at the input of the GIC.

### 3.2. Proposed Implementation of General Immittance Converter

To prove our theoretical concept in designing a series of floating fractional-order elements, we also propose possible circuit implementation, whose performance is analyzed in detail in Section 4. To implement the required GIC, the well-known OTAs are used as active elements.

The OTA, whose circuit symbol is shown in Figure 3, specified with its transcondutance $g_{m}$ is a source of current $i_{\mathrm{OUT}}$ controlled by a difference of input voltages $v_{+}$ and $v_{-}$ [50]:(10)$$i_{\mathrm{OUT}1}=i_{\mathrm{OUT}2}=g_{m}\left(v_{+}-v_{-}\right),$$
whereas $g_{m}$ may commonly be adjusted by an external dc voltage $V_{\mathrm{SET}}$ or current $I_{\mathrm{SET}}$.

In Figure 4, the novel configuration of a general immittance converter is shown. Taking into account the basic terminal relationship of OTA (Equation 10) and performing routine algebraic analysis, the input admittance is determined as:(11)$$\left[Y_{\mathrm{IN}}\right]\;=\;\left[\begin{array}{cc} 1 & -1\\ -1 & 1 \end{array}\right]\frac{\prod_{i=1}^{n/2}Y_{\left(2i\right)}}{\prod_{i=1}^{n/2}Y_{\left(2i-1\right)}}\frac{\prod_{i=1}^{n/2+2}g_{m(2i-1)}}{\prod_{i=1}^{n/2+1}g_{m(2i)}}.$$

Comparing (Equation 11) with (Equation 6), it may be observed that the proposed circuit from Figure 4 fully follows the behavior of a general immittance converter as defined in Section 3.1, whereas for the transconductance *g* it holds:(12)$$g=\frac{\prod_{i=1}^{n/2+2}g_{m(2i-1)}}{\prod_{i=1}^{n/2+1}g_{m(2i)}}.$$

The following beneficial features of the proposed general immittance converter are identified:Floating fractional-order elements are designed,Only grounded external admittances are employed,Electronic tunability of $|Y_{\mathrm{IN}}|$ is possible by proper adjustment of the transcondutances $g_{m}$ of the active elements,There is no restriction concerning matching between passive (external) or active elements.

## 4. Performance Analysis of the Proposed Immittance Converter

In theory, using the proposed OTA-based general immittance converter from Figure 4, the feasible range of the fractional order α is [−n,n], whereas *n* is generally an arbitrary even integer number.

For a more practical design of a series of fractional-order elements, let n=4. The general immittance converter from Figure 4 simplifies to a circuit as shown in Figure 5, whose input admittance according to (Equation 11) is specified as:(13)$$\left[Y_{\mathrm{IN}}\right]=\left[\begin{array}{cc} 1 & -1\\ -1 & 1 \end{array}\right]\frac{Y_{2}Y_{4}}{Y_{1}Y_{3}}\frac{g_{m1}g_{m3}g_{m5}g_{m7}}{g_{m2}g_{m4}g_{m6}}.$$

For the same reasons as already discussed in Section 3.1, assuming the external admittanaces to be suitably replaced by conductors, capacitors, and capacitive-type “seed” FOEs, the immittance converter from Figure 5 is capable of designing a series of fractional-order elements with the fractional order α in the range [−2,2]. Once the inductors and fractional inductors are used to replace one or more external admittances, the fractional order range of α will be [−4,4].

### 4.1. Properties of Used OTA

For the purpose of analysis of the real behavior of the proposed GIC from Figure 5, the OTA element designed in the 0.18 μm TSMC complementary metal-oxide semiconductor (CMOS) process as presented in [51] is used. As shown in Figure 6, the assumed OTA consists of two differential voltage summation blocks, whereas the inputs of the first one serve as differential voltage inputs of OTA and the inputs of the second summation block are used to apply the control voltage $V_{\mathrm{SET}}$. The outputs of the summation blocks are multiplied mutually and amplified with the constant *k* resulting in two output currents with the same magnitude but shifted in phase by 180 deg.

Hence, the following relation is valid for the output currents of the OTA from Figure 6:(14)$$i_{\mathrm{OUT}1}=i_{\mathrm{OUT}2}=k\cdot V_{\mathrm{SET}}\left(v_{+}-v_{-}\right),$$
where $k=2\cdot 10^{-3}\;A/V^{2}$ and defines the transmission of the block *k* in Figure 6.

In accordance with (Equation 10), the relation between $g_{m}$ and $V_{\mathrm{SET}}$ is given by:(15)$$g_{m}=k\cdot V_{\mathrm{SET}}.$$

The dependence of $g_{m}$ on $V_{\mathrm{SET}}$ of the OTA from Figure 6 obtained by Cadence simulations is given in Figure 7 (solid red line). It may be observed that (Equation 15) is valid for $V_{\mathrm{SET}}$ in the range 0 to 0.5 V and proves the possibility to electronically set $g_{m}$ between 0 and 1 mS, whereas the maximum absolute error is 0.02 mS (Figure 7; dashed blue line). Detailed analysis of the OTA and discussion of its parameters is given in [51]. Here we further aim to analyze the influence of real properties of OTAs on the overall performance of the proposed GIC.

In an ideal case the internal impedance of OTA input and output terminals is infinity. Considering a real OTA, its properties are commonly modeled by resistances and capacitances connected between each of the terminals and ground. Considering these OTA parasitic properties the proposed GIC from Figure 5 can be redrawn as seen in Figure 8. Assuming that all OTAs in the circuit are the same, the parasitic conductors G${}_{P}$ symbolize a parallel combination of the input and output internal resistances of OTA. Similarly, the parasitic capacitors C${}_{P}$ represent a parallel combination of OTA input and output internal capacitances. Based on [51], their approximate values used in this analysis are $G_{P}\approx 1/\left(346\;k\Omega \right)=2.89\;\mu$S and $C_{P}\approx 0.28$ pF. Note that the parasitic elements in the node E express the properties of twice the number of OTAs, thus their conductance and capacitance are double compared to the other parasitic elements, i.e., $2G_{P}$ and $2C_{P}$. As the overall input port of the GIC labeled as F is differential, the terminal parasitic elements *G*${}_{P}$ and *C*${}_{P}$ are connected in series here (through ground) and thus these parasitics are considered to be $G_{P}/2$ and $C_{P}/2$. If the GIC is connected as single-ended, i.e., one of its input terminals is grounded, the values of the parasitic elements of the input node should be considered to be $G_{P}$ and $C_{P}$.

### 4.2. Influence of OTA Parasitics and Optimization of GIC Performance

To solely evaluate the influence of OTA parasitic properties, as well as FOEs, resistors, and capacitors used to replace the admittances Y${}_{1}$, Y${}_{2}$, Y${}_{3}$, and Y${}_{4}$ are assumed to be ideal. For clarity, the nodes and input port, where the modeled parasitics are present are labeled by circled letters A to F in Figure 8.

#### 4.2.1. Nodes A, B, C, D

As already mentioned in Section 3.1, the external admittances Y${}_{1}$, Y${}_{2}$, Y${}_{3}$, and Y${}_{4}$ connected to these nodes are expected to be replaced by conductors (i.e., resistors), classic capacitors, or capacitive FOEs. In case of conductors, the parallel parasitic conductance $G_{P}$ is added, but it is usually very small and can be neglected. The capacitance $C_{P}$ is also in parallel and considering operational conductance of the order of milisiemens (mS), the parasitic effect of $C_{P}$ becomes significant at a very high frequency (above approx. 500 MHz), and thus can also be neglected.

On the other hand, the replacement of external admittances by capacitors or capacitive FOEs is worth analyzing. At low frequencies these elements have a very low admittance magnitude and the parasitic conductance $G_{P}$ may prevail. In case of fractor with fractional order α and fractance *F* this happens below the frequency:(16)$$\omega_{\mathrm{GP}}\approx {\left(\frac{G_{P}}{F}\right)}^{\frac{1}{\alpha }},$$
as illustrated by asymptotic admittance magnitude plot in Figure 9.

Note that (Equation 16) is also valid for a classic capacitor when considering α=1 and a capacitance equal to *F*. It follows that for a higher value of *F* correct operating range is extended to lower frequencies. Once for a specific *F* sufficiently low $\omega_{\mathrm{GP}}$ is not provided, the parasitic conductance $G_{P}$ can be reduced, e.g., by connecting in parallel a negative conductance as described in Section 4.3 in detail. Using this approach, i.e., the negative conductance, the frequency $\omega_{\mathrm{GP}}$ can theoretically be shifted to very low values. However, the demands on the accuracy of the negative conductance increase. Reducing $G_{P}$ also decreases the lower bound of the obtainable admittance magnitudes of nodes A to D.

#### 4.2.2. Node E

In case of ideal OTAs, the order of the admittance of the node E ($Y_{E}$) equals to $\alpha_{E}=-\left(\alpha_{2}+\alpha_{4}\right)$, and hence can range from 0 to −2. Thus the character of the admittance ranges from resistive through fractional inductive, inductive, fractional FDNR-II to FDNR-II. The resistive character will be not included in the analysis, as in this case the whole lower branch including OTA${}_{2}$, OTA${}_{4}$, and OTA${}_{6}$ can be omitted and replaced by a resistor and a similar conclusion as described in Section 4.2.1 is reached. Considering all other possible characters of $Y_{E}$, the OTA parasitic properties affect the circuit mainly at high frequencies, since the admittance magnitude being defined as:(17)$$Y_{E}=\frac{g_{m2}g_{m4}g_{m6}}{\omega^{\left(\alpha_{2}+\alpha_{4}\right)}F_{2}F_{4}},$$
becomes comparable with or even lower than the admittance magnitude of the parasitics $2G_{P}$ and/or $\omega 2C_{P}$, which may prevail. The situation is illustrated by the admittance magnitude asymptotic plots in Figure 10.

The black lines are defined by the admittances of the parasitic elements and their breakpoint is at the frequency $\omega_{2P}$:(18)$$\omega_{2P}=\frac{2G_{P}}{2C_{P}}=\frac{G_{P}}{C_{P}}.$$

This frequency is approximately 1.6 MHz for the above mentioned parasitics of the OTA. The red lines ($Y_{E1}$ and $Y_{E2}$) in Figure 10 show two cases of possible admittance magnitudes of the node E that are not affected by the parasitics $2G_{P}$ and $2C_{P}$ yet. When the red lines approach the black “boundary” line represented by the admittance of the parasitics, these parasitics start to take effect. The cut-off frequency of the correct operation for the first case ($Y_{E1}$) is:(19)$$\omega_{2\mathrm{GP}}\approx {\left(\frac{g_{m2}g_{m4}g_{m6}}{2G_{P}F_{2}F_{4}}\right)}^{\frac{1}{\alpha_{2}+\alpha_{4}}}.$$

Considering the second case ($Y_{E2}$), the cut-off frequency is:(20)$$\omega_{2\mathrm{CP}}\approx {\left(\frac{g_{m2}g_{m4}g_{m6}}{2C_{P}F_{2}F_{4}}\right)}^{\frac{1}{1+\alpha_{2}+\alpha_{4}}}.$$

Both these frequencies can be increased by increasing $Y_{E}$, which can be done by increasing the product $g_{m2}g_{m4}g_{m6}$ as seen in (Equation 17). The decrease of $F_{2}F_{4}$ ($F_{2}$ and/or $F_{4}$) is also possible, however this may increase the lower cut-off frequency in the nodes B and/or D, see Section 4.2.1 and (Equation 16). The orders $\alpha_{2}$ and $\alpha_{4}$ are fixed to obtain the required order of the GIC input admittance. The cut-off frequency $\omega_{2\mathrm{GP}}$ can also be increased by decreasing $2G_{P}$ using the negative conductance compensation (see Section 4.3) until $\omega_{2\mathrm{GP}}=\omega_{2P}$. Decreasing further the parasitic conductance, the relation (Equation 20) starts to hold for the cut-off frequency. Note that if the compensation circuit described in Section 4.3 with the same OTA is used to reduce $2G_{P}$, in (Equation 18) and (Equation 20) it is necessary to assume $3C_{P}$ instead of $2C_{P}$ as the compensation circuit has its own parasitic capacitance $C_{P}$.

Under certain conditions a sharp peak in the node E admittance magnitude characteristic can occur at the cut-off frequency. This happens when the cross product of the phasors of ideal $Y_{E}$, j$\omega 2C_{P}$, and $2G_{P}$ approaches zero as illustrated in Figure 11. The behavior of the GIC can be unstable in this case and it is necessary to ensure a suitable damping of the oscillations. Damping can be provided by modifying the value of $2G_{P}$ by connecting an appropriate positive or negative conductance in parallel. However, in most cases the circuit is damped by its own parasitic properties and no modification is necessary. Excessive damping is not recommended as it can lead to an exceedingly soft transition of input admittance phase in a very broad band around the cut-off frequency.

#### 4.2.3. Port F

The port F is the overall input node of the GIC and thus $Y_{F}=Y_{\mathrm{IN}}=s^{\alpha }F_{\mathrm{IN}}$. This admittance is specified as the design criterion and thus cannot be modified during optimization. Due to the limited optimization possibilities (in fact involving only changes in parasitics) it is suitable to evaluate and optimize the GIC performance in this node first. The optimization of other nodes beyond the performance of this node brings no improvement.

The fractional order α of $Y_{F}$ is in case of ideal OTAs given by (Equation 9) and ranges from −2 to 2. If α is positive, the admittance at port F is capacitive and at high frequencies it reaches high values in magnitude compared to the admittance of parasitics $G_{P}/2$ and $\omega C_{P}/2$ present at the port. Thus the parasitics do not take effect in the port F at high frequencies and the upper frequency of the GIC operation is determined primarily by the properties of node E as described in Section 4.2.2. On the other hand, at low frequencies $Y_{F}$ can reach a low magnitude comparable with the parasitic conductance $G_{P}/2$ present at port F. This is similar to the situation described in Section 4.2.1 for nodes A, B, C, and D. Since the admittance $Y_{F}$ cannot be changed as mentioned above, the only way to broaden the operation band to lower frequencies is to reduce the parasitic conductance at port F, e.g., using the compensation technique as proposed in Section 4.3. Additionally, the cut-off frequencies of the nodes A, B, C, and D should be determined and if necessary adjusted in accordance with the cut-off frequency of port F.

If α is negative, the admittance at port F is inductive and at high frequencies it can reach a low magnitude that is comparable with the admittance of parasitics $G_{P}/2$ and $\omega C_{P}/2$. The analysis is then similar to node E, see Section 4.2.2, with the difference that in this case the magnitude of $Y_{F}$ cannot be modified. Hence, the optimization can be done only by reducing $G_{P}/2$ such that it is lower than both $|Y_{\mathrm{IN}}|$ and $\omega C_{P}/2$ at frequency, where these admittances are equal, that is:(21)$$\omega_{\mathrm{CP}/2}={\left(\frac{F_{\mathrm{IN}}}{C_{P}/2}\right)}^{\frac{1}{1-\alpha }},$$
and it is the maximum operation frequency of the port F and cannot be increased. Note that when the differential compensation circuit described in Section 4.3 with the same OTA is used to reduce parasitic conductance at port F, in (Equation 21) it is necessary to assume $C_{P}$ instead of $C_{P}/2$, as again the compensation circuit has its own parasitic capacitance $C_{P}/2$. The subsequent step is verification or prospective optimization of the cut-off frequency of the node E, whereas its value specified by (Equation 19) or (Equation 20) is to be at least as high as $\omega_{\mathrm{CP}/2}$. Also note that when the GIC is connected as single-ended, the parasitics in port F should be considered with values $G_{P}$ and $C_{P}$ instead of $G_{P}/2$ and $C_{P}/2$. The single-ended variant of the compensation circuit with negative conductance can be utilized as presented in Section 4.3.

The effectiveness of the described compensation possibilities in individual nodes is demonstrated in Section 5.3, where the overall performance of the proposed GIC is discussed.

### 4.3. OTA-Based Circuit with Negative Conductance

When the parasitic conductance present in a node of the proposed GIC is to be decreased within performance optimization as described in Section 4.2, simple compensation circuits as shown in Figure 12 can be employed.

The circuit from Figure 12a is suitable for compensation of parasitic conductance at nodes A to E, since compensation conductance in the COMP terminal relative to ground is:(22)$$G_{\mathrm{COMP}}=-g_{\mathrm{mC}}+G_{P},$$
whereas the differential variant in Figure 12b can be connected to the port F and its input conductance is:(23)$$G_{\mathrm{COMP}}=-g_{\mathrm{mC}}+\frac{G_{P}}{2}.$$

We consider that the utilized OTA${}_{C}$ has the same parasitic terminal properties as the OTAs used in the proposed GIC. The conductance $G_{\mathrm{COMP}}$ can be set to an appropriate negative value by the setting of $g_{\mathrm{mC}}$. It should be again noted that when connecting the circuits from Figure 12 to a node or port, the total parasitic capacitance in the node or port increases by the parasitic capacitance of the compensation circuit which is $C_{P}$ or $C_{P}/2$ in the case of Figure 12a,b, respectively. Thus it is necessary to take this value into account in the relations containing the parasitic capacitance of the node or port being optimized.

## 5. Simulation Results

To prove the functionality of the proposed GIC and mainly to show its advantageous feature in designing a wide set of fractional-order elements using a very limited count of “seed” FOEs, the performance of the GIC was further verified by post-layout simulations in Cadence Virtuoso 6.1.6.

First, two “seed” FOEs are designed and further utilized in the proposed GIC, whereas following the recommendations from Section 4.2, the optimization steps are also verified to improve the overall performance of the GIC. Additionally, as an example, a band-pass filter is designed using the fractional-order FDNR-I with fractional order α=1.75.

### 5.1. Design of “Seed” FOEs

To obtain the set of new FOEs and their fractional-order α as listed in Table A2, the “seed” FOEs with $\alpha_{\mathrm{seed}1}=0.25$ and $\alpha_{\mathrm{seed}2}=0.0625$ are required. Due to the commercial unavailability of such FOEs, these “seed” FOEs were approximated by 7th-order Valsa topology as shown in Figure 13. The resistances and capacitances were determined using the approach described in [34] and are summarized in Table 1. Computed resistor and capacitor values are the E48 and E12 series EIA standard compliant RC values, respectively. The fractances of the two “seed” FOEs are $F_{\mathrm{seed}1}=112.3\;\mu$Fs${}^{-0.75}$ and $F_{\mathrm{seed}2}=578.9\;\mu$Fs${}^{-0.9375}$, respectively, and their admittance at central frequency of approximation 1 kHz is 1 mS.

In Figure 14 the magnitude and phase admittance frequency characteristics of the approximated “seed” FOEs are shown (solid lines) and compared with ideal “seed” FOEs (dotted lines). The absolute errors in magnitude and phase of the approximated “seed” FOEs are also depicted (dashed lines), whereas the correct operation may be observed in 4 decades, i.e., from 10 Hz to 100 kHz.

Here we should note again that the general admittances Y${}_{1}$, Y${}_{2}$, Y${}_{3}$, and Y${}_{4}$ are external and to be replaced by discrete resistors, capacitors, and/or “seed” FOEs. Hence, the accuracy of the fractional order α of the FOE being observed at the input of the GIC is determined only by the accuracy of the “seed” FOEs since the external resistors and capacitors are always characteristic with 0 deg and −90 deg phase shift, respectively. If higher accuracy of the fractional order α is required, the accuracy of $\alpha_{\mathrm{seed}}$ must also be increased by commonly increasing the order of the RC network used to approximate the “seed” FOE [34] or using a different RC network [33].

### 5.2. Simulation of the GIC

To implement the proposed GIC, the OTA cell designed in the 0.18 μm TSMC CMOS process as presented in Section 4.1 and described in detail in [51] was used. Since for our purpose we use the OTA cell as the final block, here we do not further focus in detail on its layout design, as our prime aim is the proposal of the concept designing a series of FOEs using “seed” FOEs. Those interested in issues regarding the chip layout design may refer to [52,53,54].

The overall circuit layout of the proposed GIC is shown in Figure 15. In Figure 15, the cells OTA${}_{i}$ (i=1,2…7) correspond to prime active elements of the GIC circuit as shown in Figure 5. The cells OTA${}_{C-j}$ (j=A,B…F) represent the single-ended or differential compensation circuit from Figure 12 to reduce the parasitic conductance $G_{P}$ present in the nodes A to E, or port F as labeled in Figure 8. Additionally, the block I${}_{\mathrm{BIAS}}$ is a set of current sources to bias the OTA cells. The labels Y${}_{1}$, Y${}_{2}$, Y${}_{3}$, Y${}_{4}$, and Y${}_{\mathrm{IN}}+$, Y${}_{\mathrm{IN}}-$ represent the pins, to which the external discrete elements, i.e., resistors, conductors, and/or “seed” FOEs are to be connected, or the input terminal of the GIC, respectively.

Within the simulations, next to the “seed” FOEs as described in Section 5.1, the external general admittances Y${}_{1}$, Y${}_{2}$, Y${}_{3}$, and Y${}_{4}$ of the GIC are always replaced by 1 mS admittances or 159.2 nF capacitors (as at the central frequency 1 kHz their admittance is 1 mS). The transconductances $g_{m}$ of all prime OTAs are 1 mS (i.e., $V_{\mathrm{SET}}=0.5$ V). The resulting magnitude and phase characteristics of the input admittance of the immittance converter from Figure 5 are presented in Figure 16. The black dotted lines represent the results with ideal OTAs and approximated “seed” FOEs employed. To maintain the clarity of the simulation results being displayed in Figure 16, only the α values from the range [−2,2] with the step 0.25 were selected. Based on the values of external admittances and setting of OTAs, the input admittance magnitude of the GIC is always $|Y_{\mathrm{IN}}|=1$ mS at 1 kHz.

In Figure 16 it is apparent that the input admittance magnitude and phase characteristics are affected by the OTAs parasitic properties. Most distorted are the characteristics for |α|>1, both at low and high frequencies, whereas in the magnitude characteristics (Figure 16a) peaking is evident in several cases. This peaking is caused by the resonance of the node E or port F admittance (which has a character of fractional or integer-order FDNR-I or FDNR-II) with OTA parasitic conductance. Fortunately, damping of the oscillations is always ensured by the OTA parasitic capacitance and thus the circuit is stable. However, the overall bandwidth of correct operation for the highest values of |α| reduces down to two decades only, which is two decades lower than the bandwidth of the approximated “seed” FOEs. To broaden the bandwidth of the GIC, optimization is required by following the steps as described in Section 4, which are validated in Section 5.3.

### 5.3. Optimization of GIC Performance

The influence of the OTA parasitics on the significant reduction of the operational frequency band of the newly obtained FOEs can be observed in Figure 16, mainly for |α|>1. To reach an operational bandwidth of FOEs at the input of GIC to be at least the same as it is of the “seed” FOEs (i.e., 10 Hz–100 kHz), optimization is necessary and is demonstrated on two following examples.

#### 5.3.1. Optimization Example for α=1.75

In this case, the fractional FDNR-I is obtained at the input of the GIC, i.e., port F, whose fractance is $F_{\mathrm{IN}}=22.55$ nFs${}^{0.75}$. Decreasing frequency the admittance magnitude also decreases until the parasitic conductance $G_{P}/2$ starts to prevail. This happens at a lower cut-off frequency at approximately 23.8 Hz (Figure 16a) generally determined by (Equation 16), where $G_{P}$ was substituted by $G_{P}/2$ (note that (Equation 16) is originally valid for nodes A, B, C, D where parasitic conductance $G_{P}$ is present).

Within optimization, using the circuit from Figure 12b to compensate the parasitic conductance at port F, the lower cut-off frequency is decreased down to 1 Hz to maintain a sufficient margin to frequency 10 Hz due to soft admittance phase transition (Figure 16b). To reach this new lower cut-off frequency, the input conductance $G_{\mathrm{COMP}}$ was set to −1.439μS, whereas according to (Equation 23) compensation transconductance $g_{\mathrm{mC}}$ equals to 2.885 μS.

The upper cut-off frequency is determined by the parasitics of the node E as described in Section 4.2.2. To increase this upper cut-off frequency it is necessary either to increase the product $g_{m2}g_{m4}g_{m6}$ or to decrease $F_{2}F_{4}$ (in this case capacitances $C_{2}$ and $C_{4}$ as $\alpha_{2}=\alpha_{4}=1$). Since the transconductances $g_{m}$ of all OTAs are already set to 1 mS (maximum according to Figure 7), the product $g_{m2}g_{m4}g_{m6}$ cannot be further increased. Hence, having selected the upper cut-off frequency to be 100 kHz, using (Equation 19) new capacitances $C_{2}$ and $C_{4}$ (considering them equal) were determined to be 20.9 nF. Note that here the margin from the required 100 kHz was not considered, as the damping in the node E is low and the admittance phase shows the transition in a narrow band. Moreover, the excessive increase of the upper cut-off frequency in node E would lead to lower capacitances $C_{2}$ and $C_{4}$ and undesirable deterioration of the cut-off frequency in nodes B and D.

Within the optimization of the upper cut-off frequency, the ratio $g_{m2}g_{m4}g_{m6}/\left(F_{2}F_{4}\right)$ was increased. Hence, to keep the original value of the input fractance $F_{\mathrm{IN}}$ unaffected, according to general formula (Equation 13), the ratio $G_{1}F_{\mathrm{seed}1}/\left(g_{m1}g_{m3}g_{m5}g_{m7}\right)$ must decrease. As again the transconductances $g_{m}$ of all OTAs are already set to their maximum values (i.e., 1 mS) and the “seed” FOE is not expected to be modified, the only possibility is to decrease $G_{1}$ to 17.3 μS.

#### 5.3.2. Optimization Example for α=−1.75

For this case, the fractional FDNR-II with fractance 4434 Fs${}^{-2.75}$ is obtained at the input of the GIC. The admittance magnitude decreases with increasing frequency, where the parasitics at port F define the upper cut-off frequency of approximately 42 kHz (Figure 16a) generally determined by (Equation 16), where again $G_{P}$ was substituted by $G_{P}/2$. The only solution to increase the upper cut-off frequency is to reduce the parasitic conductance of the port F by using the compensation circuit from Figure 12b. In this case it is possible to decrease the port F parasitic conductance almost to zero, thus the transconductance of the compensation circuit is set slightly lower than $G_{P}$, i.e., $g_{\mathrm{mC}}=2.888$ μS.

To reduce the lower cut-off frequency, it is necessary to increase capacitances $C_{1}$ and $C_{3}$ in the nodes A and C according to (Equation 16). The optimized lower cut-off frequency is set to 1 Hz to have again sufficient margin to 10 Hz due to soft phase transition. Hence, the new value of capacitances $C_{1}$ and $C_{3}$ is 460 nF.

Within the optimization of the lower cut-off frequency the product $C_{1}C_{3}$ was increased. Hence, the ratio $g_{m2}g_{m4}g_{m6}/\left(G_{2}F_{\mathrm{seed}1}g_{m1}g_{m3}g_{m5}g_{m7}\right)$ must decrease according to general formula (Equation 13) to keep the original value of the input fractance $F_{\mathrm{IN}}$ unchanged. For this purpose, the transconductances $g_{m2}$, $g_{m4}$, and $g_{m6}$ were set to 0.493 mS, whereas $g_{m1}$, $g_{m3}$, $g_{m5}$, $g_{m7}$, $G_{2}$, and mainly $F_{\mathrm{seed}1}$ are kept the same. As transconductances $g_{m2}$, $g_{m4}$, and $g_{m6}$ were changed, it is necessary to check the upper cut-off frequency of the node E if it is large enough. According to (Equation 20) the value of $f_{2\mathrm{CP}}$ is 4.2 MHz, which is much more than the required upper cut-off frequency of 100 kHz. Hence no further optimization is needed.

For the both optimized examples as described in Section 5.3.1 and Section 5.3.2, the resulting admittance magnitude and phase frequency characteristics are shown in Figure 17 along with the characteristics of the non-optimized GIC taken from Figure 16. It is evident that the optimized circuit provides a higher frequency bandwidth of the admittance characteristics covering the required 4 decades. The fractional FDNR-I (blue lines) reaches an upper cut-off frequency almost equal 100 kHz as considered during the optimization. The lower cut-off frequency reached approximately 5 Hz, which is higher than the projected value of 1 Hz, however, here the GIC function is affected by parasitics of multiple nodes and also the “seed” FOE shows a higher error (see Figure 14). The fractional FDNR-II (red lines) has also been optimized successfully. Its upper cut-off frequency is around 100 kHz and lower cut-off frequency is 0.9 Hz. Additionally, as seen from Figure 17a, the dynamic range of the admittance magnitude has also increased thanks to the optimization.

### 5.4. Fractional Band-Pass Filter Design

To also show the practical utilization of the proposed GIC and the fractional-order element that are being obtained at its input, a fractional band-pass filter as presented in Figure 18 is designed, as an example.

The transfer function of the filter from Figure 18 is determined as:(24)$$TF_{FBP}\left(s\right)=\frac{as^{\alpha -1}}{s^{\alpha }+as^{\alpha -1}+b},$$
where a=1/(CR) and b=1/(FR), whereas *F* is the fractance of fractional FDNR-I (FOFDNR-I) with its fractional order being in the range 1<α<2.

According to (Equation 24), the band-pass filter features stop-band attenuation of +20α dB/dec and −20 dB/dec for frequencies lower and higher than the pole frequency, respectively.

For Butterworth approximation of fractional-order band-pass filters, based on [55] the coefficients *a* and *b* are determined as:(25)$$a=\omega_{0}\left(0.7141-1.1632\alpha +0.7516\alpha^{2}\right),$$
and
(26)$$b=\omega_{0}^{\alpha }\left(1.5464-1.3562\alpha +0.5357\alpha^{2}\right),$$
where $\omega_{0}$ is the angular pole frequency of the filter.

Assuming the FOFDNR-I with its fractance F=22.55 nFs${}^{0.75}$ and fractional order α=1.75, as is obtained at the input of GIC and using (Equation 24)–(Equation 26), the values of resistor and capacitor of the filter from Figure 18 can be determined as R=21.88
Ω and C=742 nF for pole frequency $f_{0}=10$ kHz.

The magnitude and phase frequency responses of the band-pass filter reached by simulations are shown in Figure 19 and compared to ideal behavior. Within simulations, the FOFDNR-I was assumed to be implemented prior and after the GIC performance optimization as discussed in Section 5.3.1. From the simulation results it can be seen that the filter follows the ideal behavior very well, mainly for the optimized design of the required FOFDNR-I (solid lines). The proper behavior of the filter may be observed in four decades, which corresponds to optimized GIC performance and even the bandwidth of the initial “seed” FOEs. The most significant differences can be seen in the results of the non-optimized circuit above pole frequency, where a greater slope of attenuation was achieved. This is caused by parasitics in node E of the GIC used, which manifest themselves at a frequency of 10 kHz, as described before in Section 5.3.1.

## 6. Conclusions

In this paper we presented the concept of an efficient design of fractional-order element series in fractional order α using a very limited count of initial FOEs, here referred to as “seed” FOEs. The proposed concept is powerful and significantly helps to overcome the current obstacle of commercial unavailability of FOEs and was based on the utilization of general immittance converter, in addition a novel general OTA-based implementation was also proposed. To show the advantageous features of the proposed concept, as an example two “seed” FOEs with fractional orders 0.25 and 0.0625 were implemented to design a series of new 51 FOEs with unique fractional order in the range [−2,2]. The “seed” FOEs were approximated using the Valsa RC network in four decades featuring a very low absolute error. Comprehensive analysis of the designed circuit was given to enable its performance optimization. Using OTAs designed in 0.18 μm TSMC CMOS technology, Cadence Virtuoso post-layout simulation results were presented which prove the operability of the proposed GIC, whereas the performance optimization was also shown on two examples to extend the operational frequency range. Finally, a fractional-order band-pass filter was also designed, which successfully utilizes the floating fractional FDNR-I with its fractional order α=1.75.

## Figures and Tables

**Figure 1 sensors-21-01203-f001:**
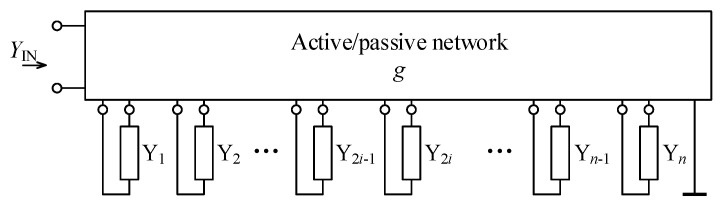
View on general immittance converter as a function block.

**Figure 2 sensors-21-01203-f002:**
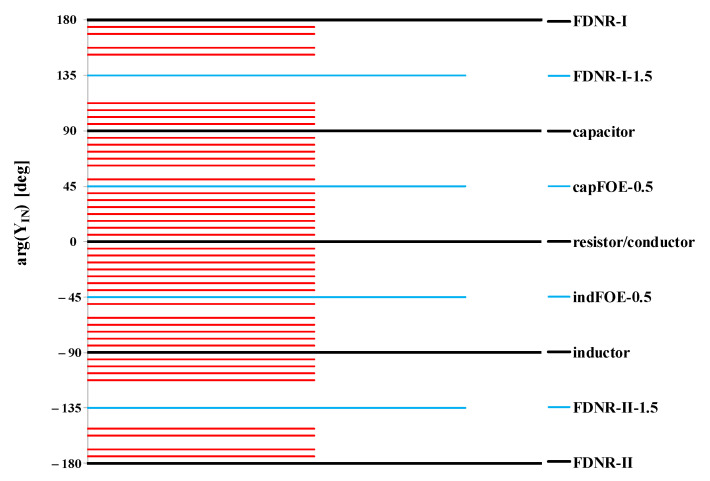
Feasible phase angles of $Y_{\mathrm{IN}}$ (Equation 8) using up to two seed fractional-order elements (FOEs) with $\alpha_{\mathrm{seed}1}=0.25$ and $\alpha_{\mathrm{seed}2}=0.0625$.

**Figure 3 sensors-21-01203-f003:**
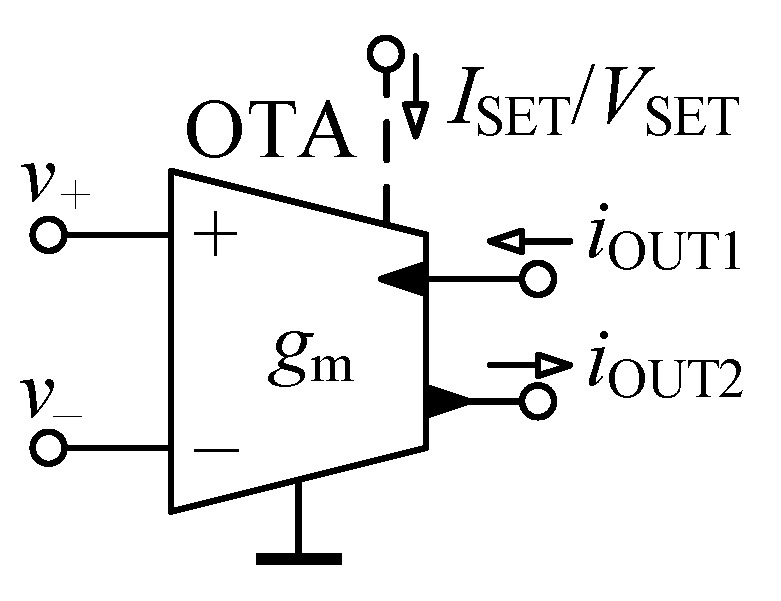
Schematic symbol of operational transconductance amplifier (OTA).

**Figure 4 sensors-21-01203-f004:**
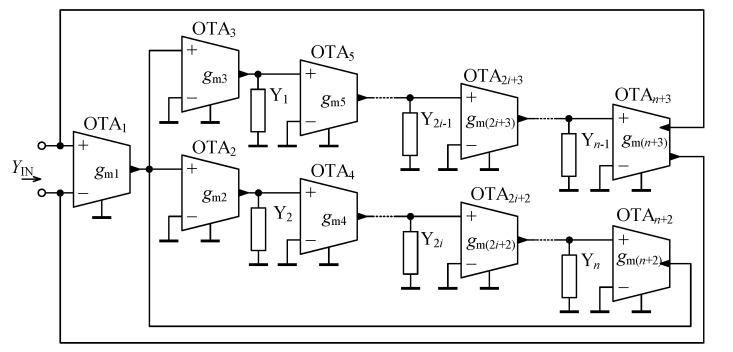
Proposed OTA-based general immittance converter.

**Figure 5 sensors-21-01203-f005:**
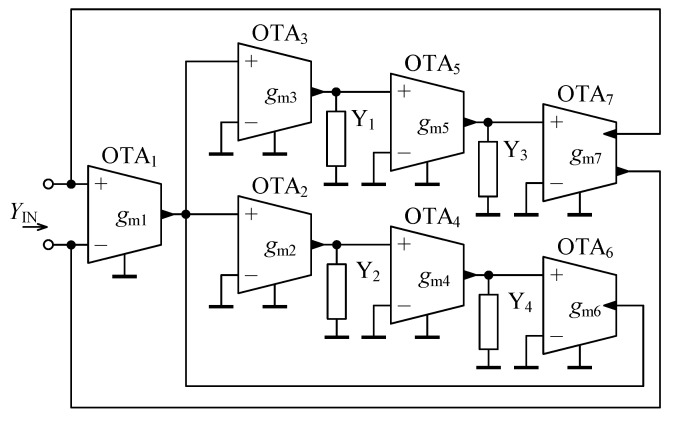
Proposed OTA-based general immittance converter for n=4.

**Figure 6 sensors-21-01203-f006:**
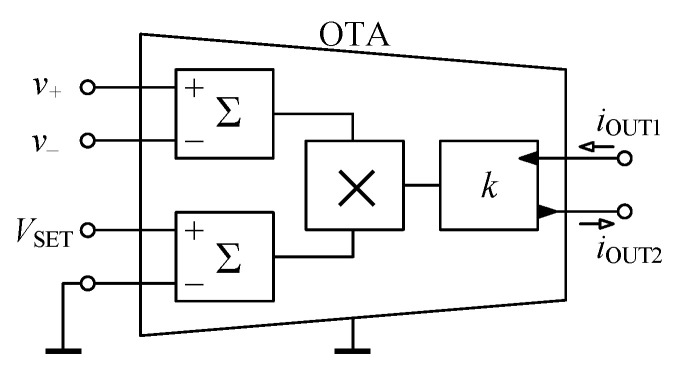
Behavioral structure of the used OTA.

**Figure 7 sensors-21-01203-f007:**
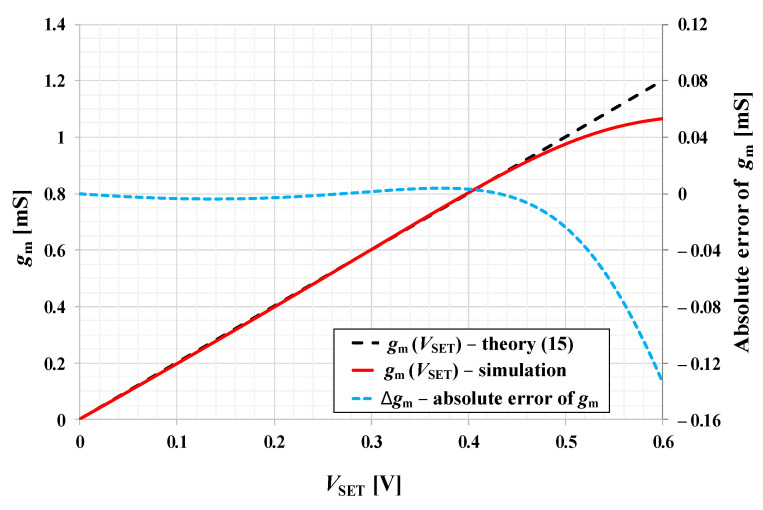
Dependence of the transconductance $g_{m}$ on $V_{\mathrm{SET}}$ (solid red line) of the OTA element and its absolute error (dashed blue line).

**Figure 8 sensors-21-01203-f008:**
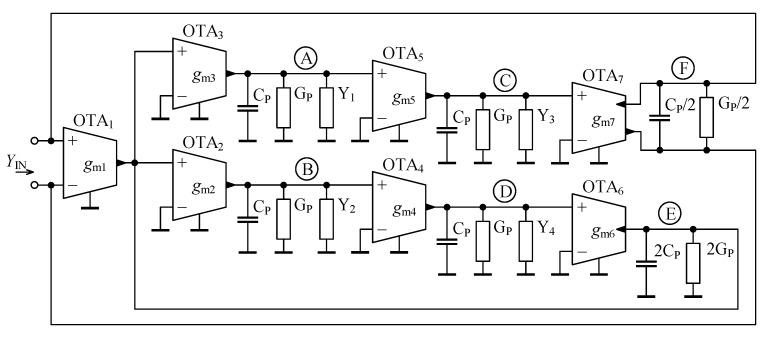
Proposed general immittance converter with OTA parasitic properties.

**Figure 9 sensors-21-01203-f009:**
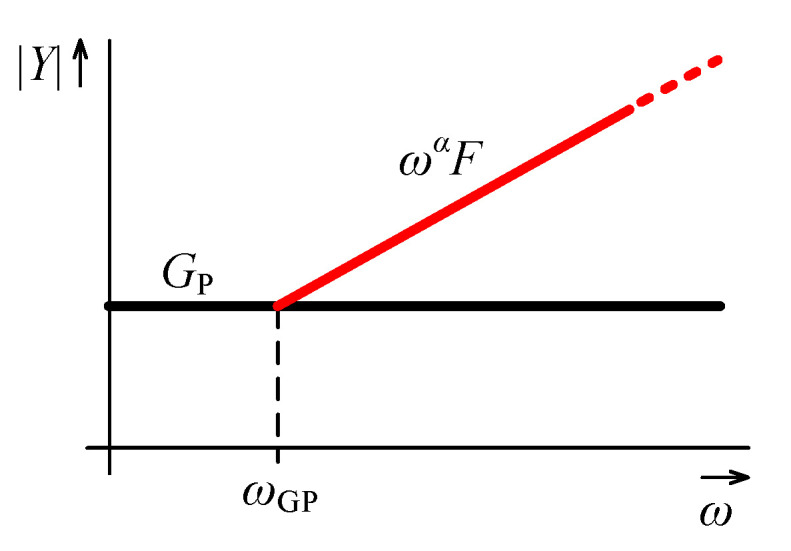
Magnitude frequency characteristics of the working and parasitic admittances of the nodes A to D.

**Figure 10 sensors-21-01203-f010:**
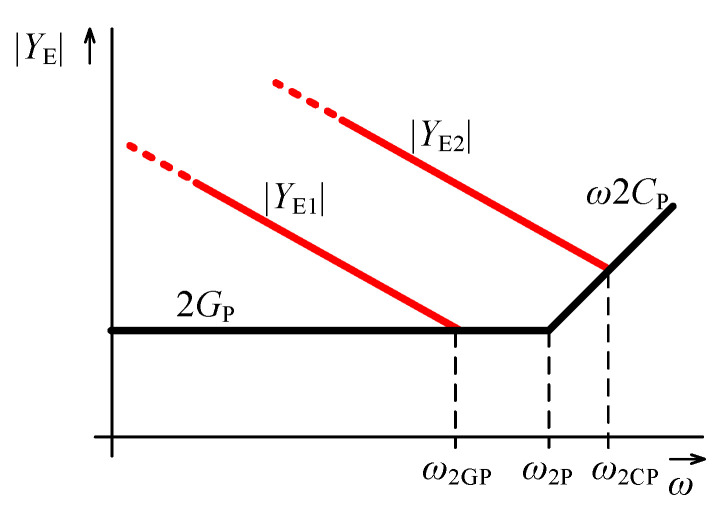
Magnitude frequency characteristics of the working (red lines) and parasitic (black lines) admittances of the node E.

**Figure 11 sensors-21-01203-f011:**
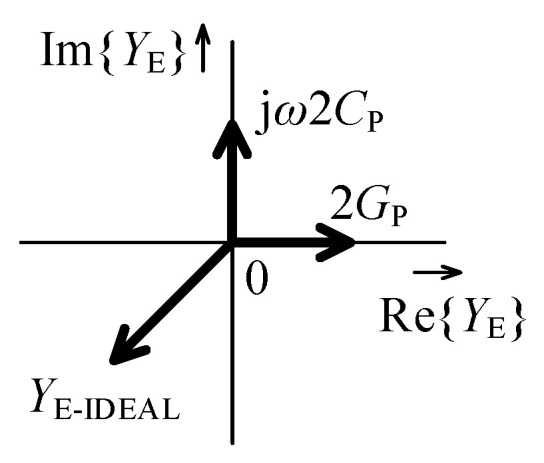
Phasor diagram resulting in zero admittance of node E.

**Figure 12 sensors-21-01203-f012:**
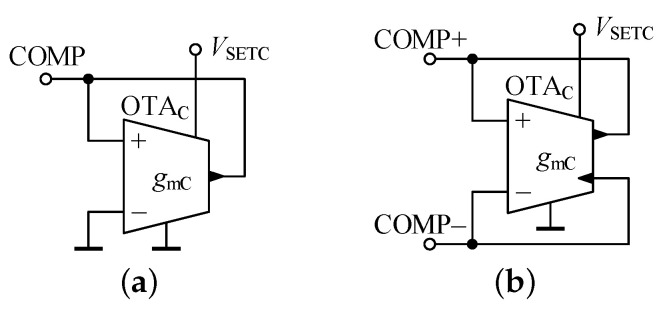
Circuit with negative conductance (**a**) between terminal COMP and ground, (**b**) between terminals COMP+ and COMP−.

**Figure 13 sensors-21-01203-f013:**
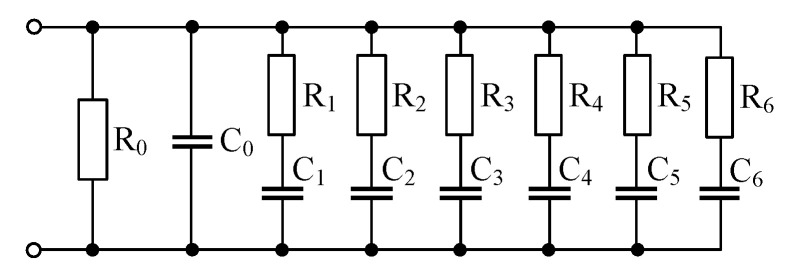
7th-order Valsa RC network to approximate “seed” FOE.

**Figure 14 sensors-21-01203-f014:**
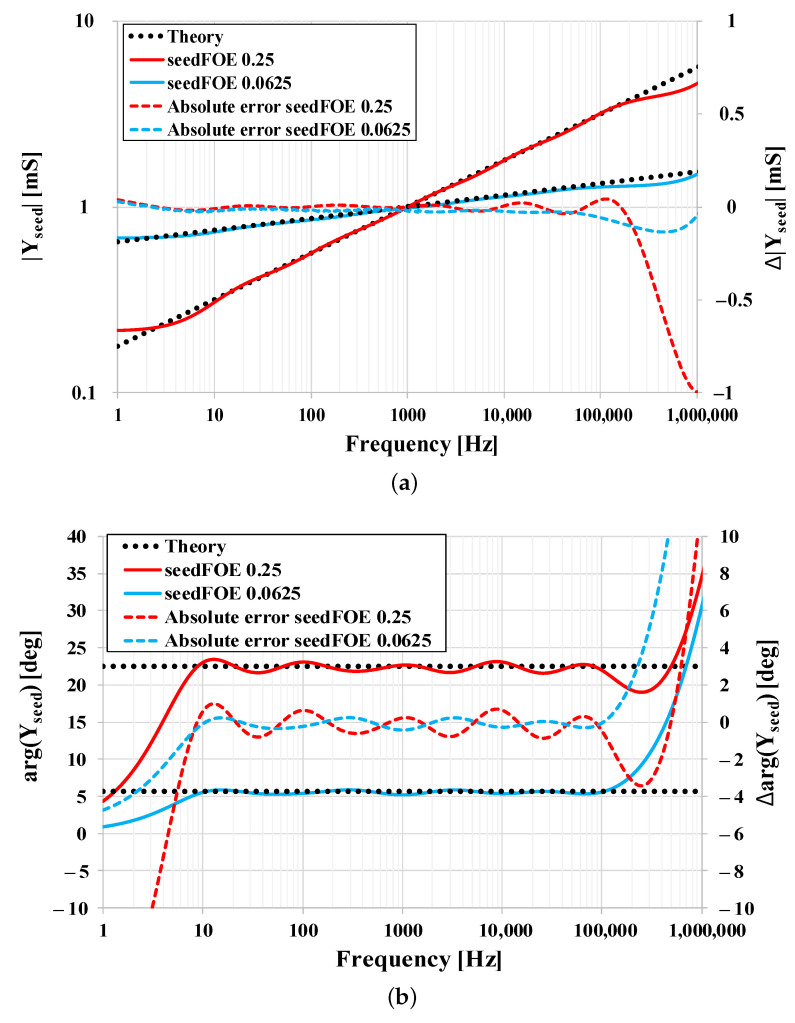
Simulation results of designed “seed” FOEs with central frequency 1 kHz: (**a**) Magnitude responses and (**b**) phase responses.

**Figure 15 sensors-21-01203-f015:**
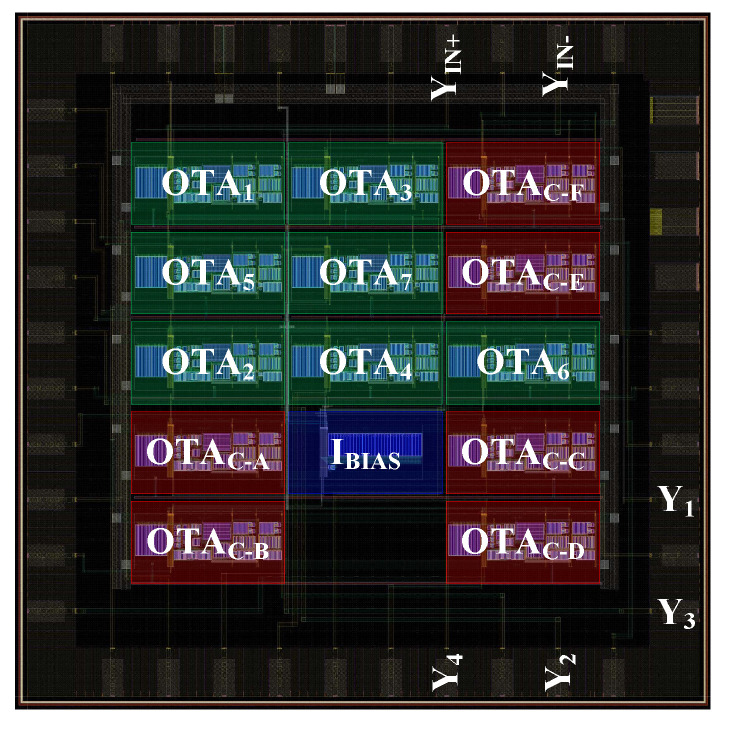
Circuit layout of the proposed general immittance converter (GIC).

**Figure 16 sensors-21-01203-f016:**
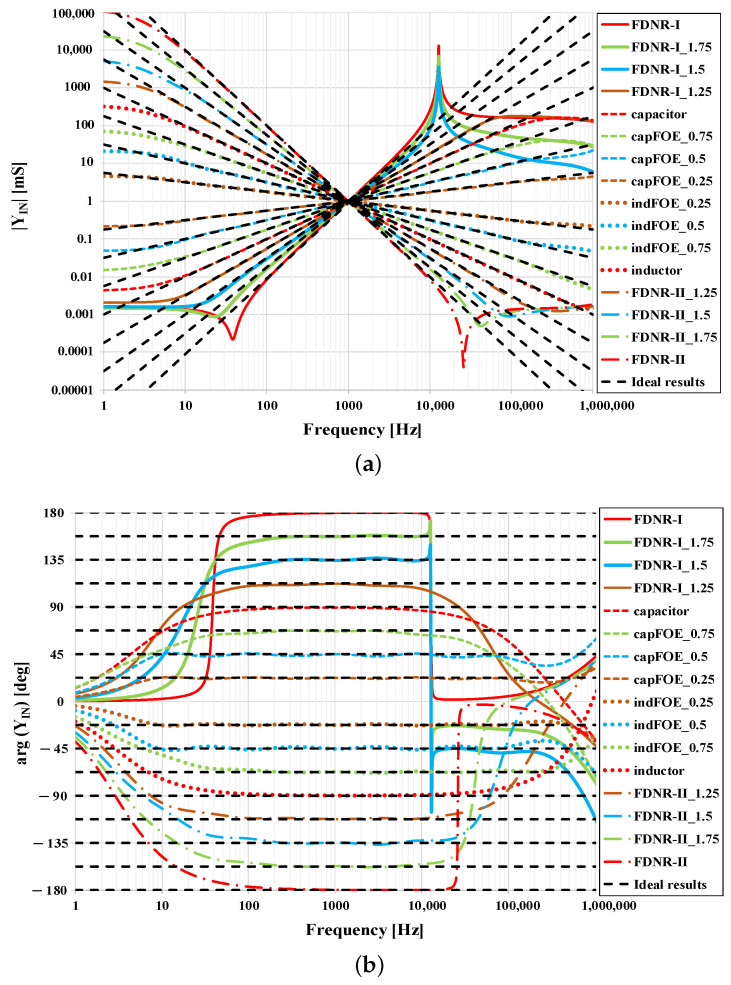
Simulation results of proposed GIC with OTA parasitics: (**a**) Magnitude responses and (**b**) phase responses.

**Figure 17 sensors-21-01203-f017:**
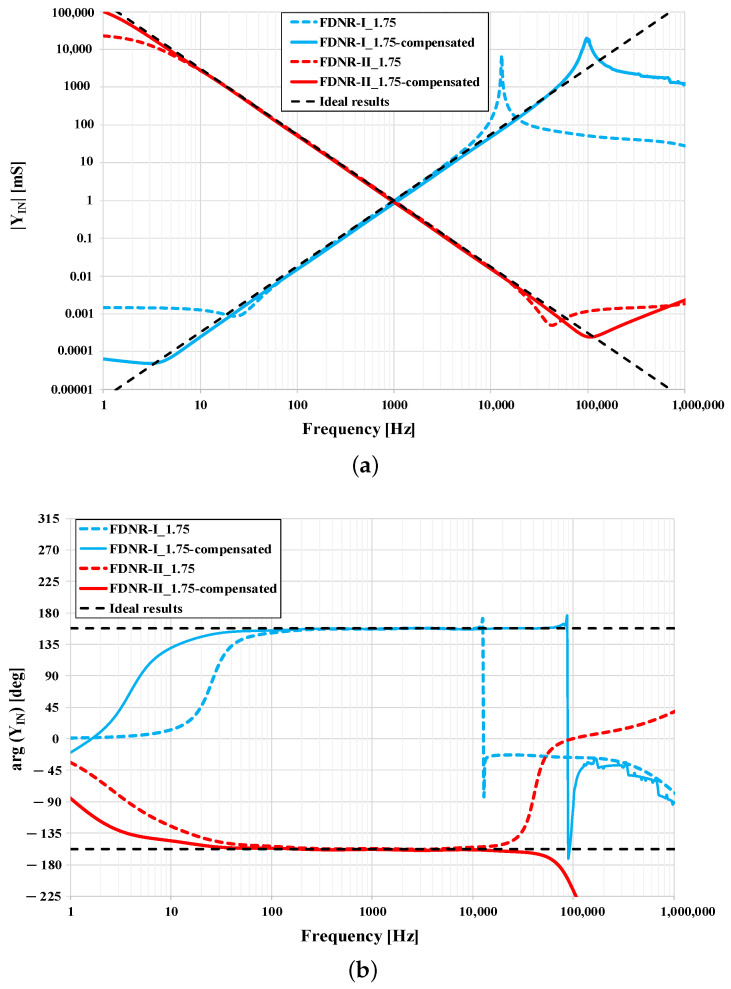
Simulation results of proposed GIC with compensated OTA parasitics: (**a**) Magnitude responses and (**b**) phase responses.

**Figure 18 sensors-21-01203-f018:**
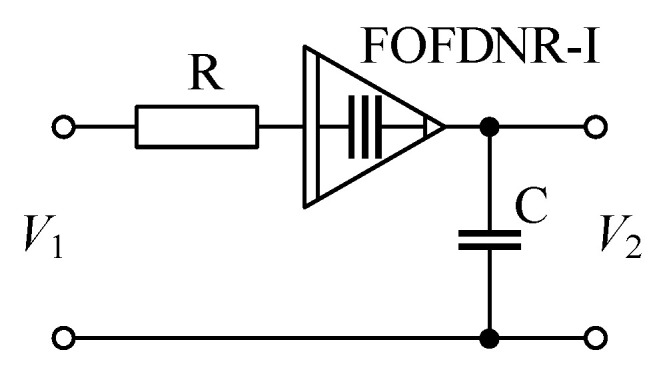
Passive band-pass filter using fractional FDNR-I element.

**Figure 19 sensors-21-01203-f019:**
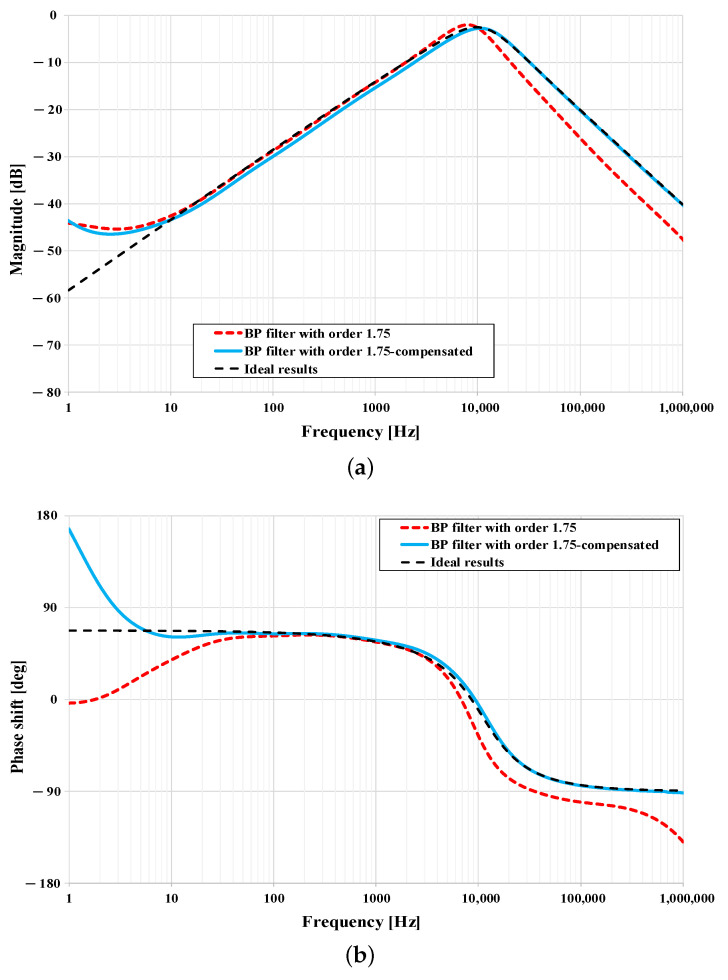
Simulation results of fractional band-pass filter from Figure 18: (**a**) Magnitude responses, and (**b**) phase responses.

**Table 1 sensors-21-01203-t001:** Resistances and capacitances in the network from Figure 13 ($\alpha_{\mathrm{seed}1}=0.25$; $F_{\mathrm{seed}1}=112.3\;\mu {\mathrm{Fs}}^{-0.75}$ and $\alpha_{\mathrm{seed}2}=0.0625$; $F_{\mathrm{seed}2}=578.9\;\mu {\mathrm{Fs}}^{-0.9375}$).

	$\alpha_{\mathrm{seed}1}=0.25$	$\alpha_{\mathrm{seed}2}=0.0625$
$R_{0}$ (kΩ)	4.64	1.47
$R_{1}$ (kΩ)	5.11	17.8
$R_{2}$ (kΩ)	4.02	13.3
$R_{3}$ (kΩ)	6.81	12.0
$R_{4}$ (kΩ)	1.15	8.20
$R_{5}$ (kΩ)	2.20	8.20
$R_{6}$ (kΩ)	0.59	6.81
$C_{0}$ (nF)	0.39	0.12
$C_{1}$ (nF)	2200	120
$C_{2}$ (nF)	330	0.47
$C_{3}$ (nF)	56	0.39
$C_{4}$ (nF)	12	1500
$C_{5}$ (nF)	47	56
$C_{6}$ (nF)	2.7	6.8

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
