# Peer review of "On Systematic Design of Fractional-Order Element Series"

_sensors, 2021, doi:10.3390/s21041203_

Round 1

Reviewer 1 Report

This paper proposed a concept for efficient design of a series of floating fractional-order elements. Overall, it is well written and prepared. I think it can be accepted. Just one minor suggestion, Key words selection can be improved in my opinion.

Author Response

Thank you to anonymous revivers to provide their feedback and comments to our manuscript. Please see attached file giving the comments of al reviweres and our response on manuscript revision.

Reviewer 2 Report

The paper needs revision. Please see the attached file.

Author Response

Thank you to provide their feedback and comments to our manuscript. Attached please find the file giving all comments also of other reviewers and our response to changed made in the manuscript.

Reviewer 3 Report

Overall, the simulations and contents are interesting. Therefore, I recommend that this manuscript to accept after minor changes.

1. It would be nice if there was a supplementary explanation for the use of fractional order and the addition of a recent paper in the introduction.

2. I think that the dot does not necessary in eq. (6). It would be nice to erase it like the expression in eq. (8).

3. Figure 7: The explanation and figure need to be supplemented because there are vague parts.

Author Response

(The authors gave the same response as above.)
